# Highly explosive basaltic eruptions driven by CO$_2$ exsolution

Chelsea M. Allison [1,2,3], Kurt Roggensack[1] & Amanda B. Clarke[1,4]

The most explosive basaltic scoria cone eruption yet documented (>20 km high plumes) occurred at Sunset Crater (Arizona) ca. 1085 AD by undetermined eruptive mechanisms. We present melt inclusion analysis, including bubble contents by Raman spectroscopy, yielding high total CO$_2$ (approaching 6000 ppm) and S (~2000 ppm) with moderate H$_2$O (~1.25 wt%). Two groups of melt inclusions are evident, classified by bubble vol%. Modeling of post-entrapment modification indicates that the group with larger bubbles formed as a result of heterogeneous entrapment of melt and exsolved CO$_2$ and provides evidence for an exsolved CO$_2$ phase at magma storage depths of ~15 km. We argue that this exsolved CO$_2$ phase played a critical role in driving this explosive eruption, possibly analogous to H$_2$O exsolution driving silicic caldera-forming eruptions. Because of their distinct gas compositions relative to silicic magmas (high S and CO$_2$), even modest volume explosive basaltic eruptions could impact the atmosphere.

[1] School of Earth and Space Exploration, Arizona State University, PO Box 876004Tempe, AZ 85287-6004, USA. [2] Department of Earth and Atmospheric Sciences, Cornell University, Snee Hall, Room 2122, Ithaca, NY 14853, USA. [3] City College of New York, City University of New York, 160 Convent Avenue, New York, NY 10031, USA. [4] Istituto Nazionale di Geofisica e Vulcanologia, Sezione di Pisa Via della Faggiola, 32, 56126 Pisa, Italy. ✉email: cmalliso@asu.edu

**S**unset Crater, a small, monogenetic scoria cone composed of alkali basalt in northern Arizona, was the source of multiple sub-Plinian basaltic events (plumes >20 km high[1]) ~1000 years ago (1085 AD[2]). It has recently been documented as the most explosive scoria cone eruption identified on Earth to date[1], but the driving mechanism of such highly explosive basaltic eruptions is unclear[3]. Some recent studies[4–6] have focused on rapid microlite crystallization and corresponding rheology changes to explain explosive behavior in basaltic magmas. In viscous, silicic systems, large explosive eruptions can be triggered when the magmatic system becomes overpressurized by processes such as crystallization-induced volatile exsolution[7], new magma injection[8], or external changes in the pressure regime[9]. These processes, however, may be less applicable to basaltic magmas of much lower viscosity. Nevertheless, explosive volcanic eruptions can inject ash and volatiles into the stratosphere with the potential to impact global climate[10]. Aside from the notable exceptions of basaltic eruptions of significant volume such as Laki (Iceland)[11] or large igneous provinces[12], the impact of basaltic volcanism on the global atmospheric system is largely unknown.

Magmatic volatile content is an integral part of the interpretation of an eruption but is difficult to measure because volatiles exsolve and escape as magma ascends and depressurizes. The best-preserved pre-eruptive concentration of magmatic volatiles is found within melt inclusions (MIs) that are trapped inside of growing crystals at depth within the magma plumbing system. Because MIs are isolated from the surrounding magma by their crystal host, they theoretically serve as a record of pre-eruptive magma composition and volatile content at the time and location of their entrapment[13]. MIs, however, are susceptible to modification from post-entrapment crystallization and shrinkage[14] during ascent, eruption, and quench at the surface[13], which often results in significant $CO_2$ loss to a bubble within the MI (Fig. 1a)[15–21]. MI bubbles may not develop solely post-entrapment, however, as they can also originate as a co-entrapped exsolved phase[22,23].

One approach to determine MI bubble contents is in situ measurement by Raman spectroscopy[17,18,20,24]. A number of recent studies using Raman and other methods have found up to 90% of the total MI $CO_2$ sequestered in the bubble, demonstrating the importance of MI bubbles in calculating magmatic volatile budgets[17–21]. Notably, these previous studies of MI bubbles examined samples with relatively low $CO_2$ content in the MI glass, generally ~100–200 ppm but up to 1500 ppm in rare samples[17–20].

Here we present an investigation of the total volatile budget of the basaltic sub-Plinian eruption of Sunset Crater by MI analysis, including Raman spectroscopic measurements of MI bubbles. We model the size of MI bubbles that can develop post-entrapment and demonstrate that an exsolved $CO_2$ phase existed in the magma at ~15 km depth. We compare magmatic volatiles at Sunset Crater to those in explosive caldera-forming silicic eruptions such as the Bishop Tuff to highlight differences in their abundance and composition. This comparison suggests that the exsolved $CO_2$ phase is a critical pre-eruptive condition that drives highly explosive basaltic eruptions. Furthermore, we constrain the total stratospheric injection of multiple volatile species by the Sunset Crater eruption and propose that basaltic eruptions, including small scoria cones, may be an overlooked source of atmospheric aerosol loading.

## Results and discussion

**MI and bubble compositions**. Analysis of MIs reveals preeruptive properties of the Sunset Crater magma. MIs are hosted in minimally normally zoned Fo ~82–85 olivine phenocrysts sampled from tephra from the first of several sub-Plinian phases of the eruption (phase 3). The MIs are largely homogeneous in major element composition corrected for 4–11% post-entrapment crystallization (see Supplementary Material). Bubbles are ubiquitous in MIs from all phases of the Sunset Crater eruption, with most MIs containing a single bubble that, in the samples analyzed here, ranges in size from 0.82 vol% to 3.26 vol% of the host MI (Fig. 1b, c). Throughout this section, we present data demonstrating that MIs should be classified into two groups based on bubble vol%. MIs with bubbles <2.5 vol% are hereafter referred to as "Group I" (black filled symbols in Figs. 2–4; example MI in Fig. 1b) and those with bubbles >2.5 vol% as "Group II" (open symbols in Figs. 2 and 3; cyan symbols in Fig. 4a, b; example MI in Fig. 1c). Total $CO_2$ contents, accounting for both the MI bubble (from Raman spectroscopy) and MI glass (from Fourier transform infrared spectroscopy (FTIR)), encompass a wide range from 2664 to 5591 ppm, with an average value of 4268 ppm (Fig. 2a and Supplementary Material). Group II MIs (bubbles >2.5 vol%) have the highest total $CO_2$ contents (>4000 ppm; open symbols in Fig. 2a). S and Cl contents measured for a different subset of phase 3 MIs are ~2000 and ~425 ppm, respectively, and show minimal variability across all samples (see Supplementary Material).

There are two different mechanisms that could produce a set of MIs with a wide range of volatile contents as observed in the Sunset Crater samples. One possible explanation is that the magma is volatile undersaturated, and so as crystallization proceeds, volatile elements that are incompatible in phenocryst phases will concentrate in the magma. The alternative explanation is that the MI volatiles record a degassing path as a volatile-saturated magma ascends and depressurizes. The data presented here show that the total $CO_2$ content generally decreases with decreasing host Fo content in these samples (Fig. 2b). This relationship implies that the magma is volatile saturated when olivine is crystallizing because $CO_2$ exsolves from the magma as crystallization proceeds and Fo content decreases. However, these

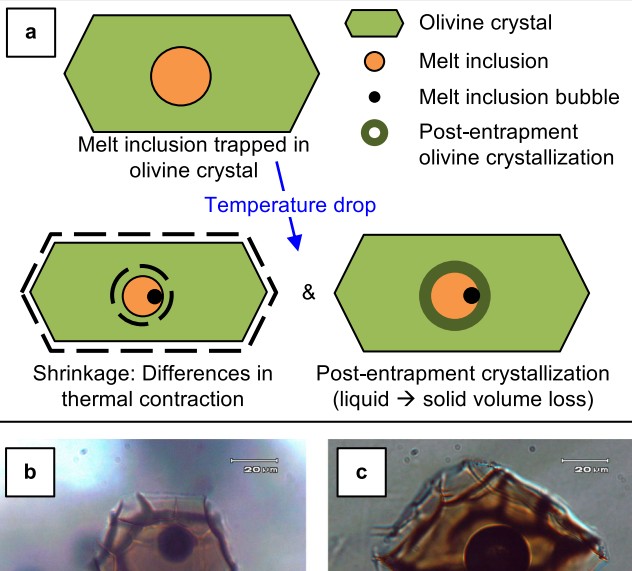

**Fig. 1 MI bubble formation and representative Sunset Crater MIs.**
**a** Diagram showing post-entrapment formation mechanisms of MI bubbles. **b**, **c** Photomicrographs of example MIs; scale bar is 20 μm. **b** MI from Group I (<2.5 vol% bubble). **c** MI from Group II (>2.5 vol% bubble).

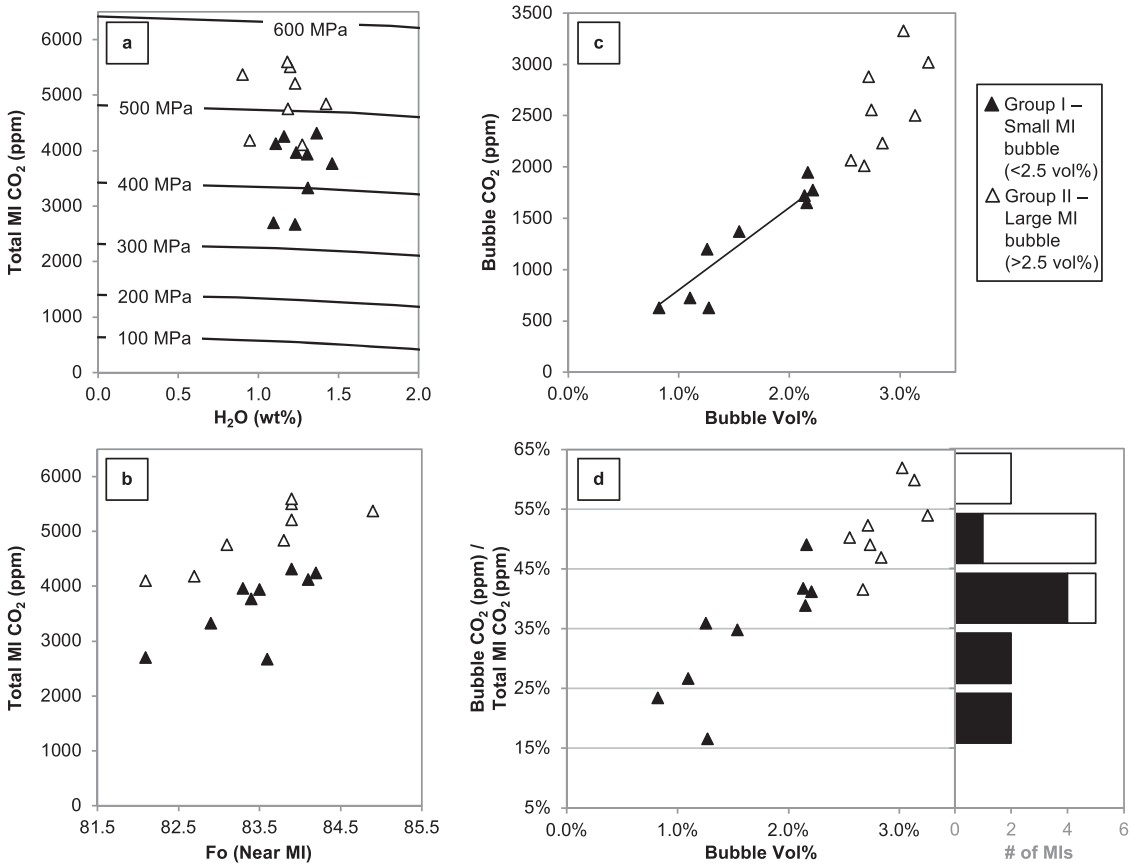

**Fig. 2 CO₂ contents of Sunset Crater MIs and bubbles.** Filled symbols are MIs with bubbles <2.5 vol% (Group I) and open symbols are MIs with bubbles >2.5 vol% (Group II). **a** $H_2O$–$CO_2$ of MIs, including all $CO_2$ from MI bubble. Curves are isobars from ref. [32]. **b** Total MI $CO_2$ vs. Fo content of the olivine host measured next to the MI. **c** Bubble $CO_2$ content vs. bubble vol%; linear trend is fit through samples with bubbles <2.5 vol%. **d** Percentage of total MI $CO_2$ content contained in MI bubble vs. bubble vol%. Data are also shown as a histogram of the percentage of total MI $CO_2$ contained in the bubble. Error from FTIR analysis is 10% for $H_2O$ and 5% for $CO_2$, while the error in the total $CO_2$ content from bubble calculations is 5%. Error in MI bubble vol% is 15% (i.e., 0.3 vol% error for a 2 vol% bubble). See "Methods" for details on error estimates.

olivine data also show two roughly parallel but distinct trends: Group II MIs (>2.5 vol% bubbles) show the same decrease in $CO_2$ with lower Fo, but offset to higher total $CO_2$ than Group I samples (Fig. 2b). So, while the Sunset Crater magma was likely volatile saturated and degassed as it ascended, this mechanism does not explain why there are two trends in Fig. 2b nor provide justification for the highest $CO_2$ contents being restricted to MIs with larger (>2.5 vol%) bubbles.

Further observations also support the division of MIs into two distinct groups by bubble vol%. Bubbles in Group II samples are proportionally larger by vol% than those in Group I, and they also typically have larger diameters, with average bubble diameters of 26 µm for Group II vs. 20 µm for Group I (see Supplementary Material). Additionally, Group I MIs define a relatively linear trend between $CO_2$ concentration in the bubble and bubble vol% (Fig. 2c), while some Group II MIs deviate from this relationship. The two groups are also distinguished by the percentage of the total MI $CO_2$ that is contained in the bubble (Fig. 2d); most Group I MIs contain <40% of their total $CO_2$ content in their bubbles, whereas all Group II MI bubbles contain >40% of the total MI $CO_2$. These results suggest that the bubbles in MIs from each of the two groups may have formed by different processes.

**Bubble growth modeling.** The two primary mechanisms of MI bubble formation include differential shrinkage of the MI and crystal host as well as crystallization at the MI–crystal interface (Fig. 1a). Shrinkage occurs because the host crystal is relatively

incompressible compared to the MI and thus the MI shrinks more than the crystal during cooling[14,25], resulting in pressure loss within the MI. Post-entrapment crystallization involves diffusion of elements from the MI into a denser crystal phase, decreasing the MI volume within its cavity and thus decreasing pressure in the MI[16,26].

Bubbles in MIs form and grow in two stages: in the subsurface due to small degrees of pre-eruptive cooling (early stage) and during rapid cooling upon eruption into the atmosphere until quench (late stage). The cooling rate of the magma during early-stage bubble growth is typically slow enough such that both post-entrapment crystallization and shrinkage occur. Additionally, because $CO_2$ solubility is very strongly pressure dependent[27], the decrease in pressure associated with these post-entrapment modifications will cause $CO_2$ to exsolve into the bubble that forms. On the other hand, in late-stage (syn-eruptive) growth, especially in explosive eruptions, cooling is extremely rapid and $CO_2$ does not have time to diffuse from the MI into the bubble[15,28]. In fact, cooling during late-stage growth is rapid enough that post-entrapment crystallization is also kinetically inhibited[20], but the bubble volume does continue to increase syn-eruption due to the shrinkage process.

The size of MI bubbles that can be generated due to post-entrapment modification processes during early- (pre-eruption) and late-stage (syn-eruption) cooling can be modeled from properties of the melt and host phenocryst at different temperatures[20]. Bubble formation during early-stage shrinkage

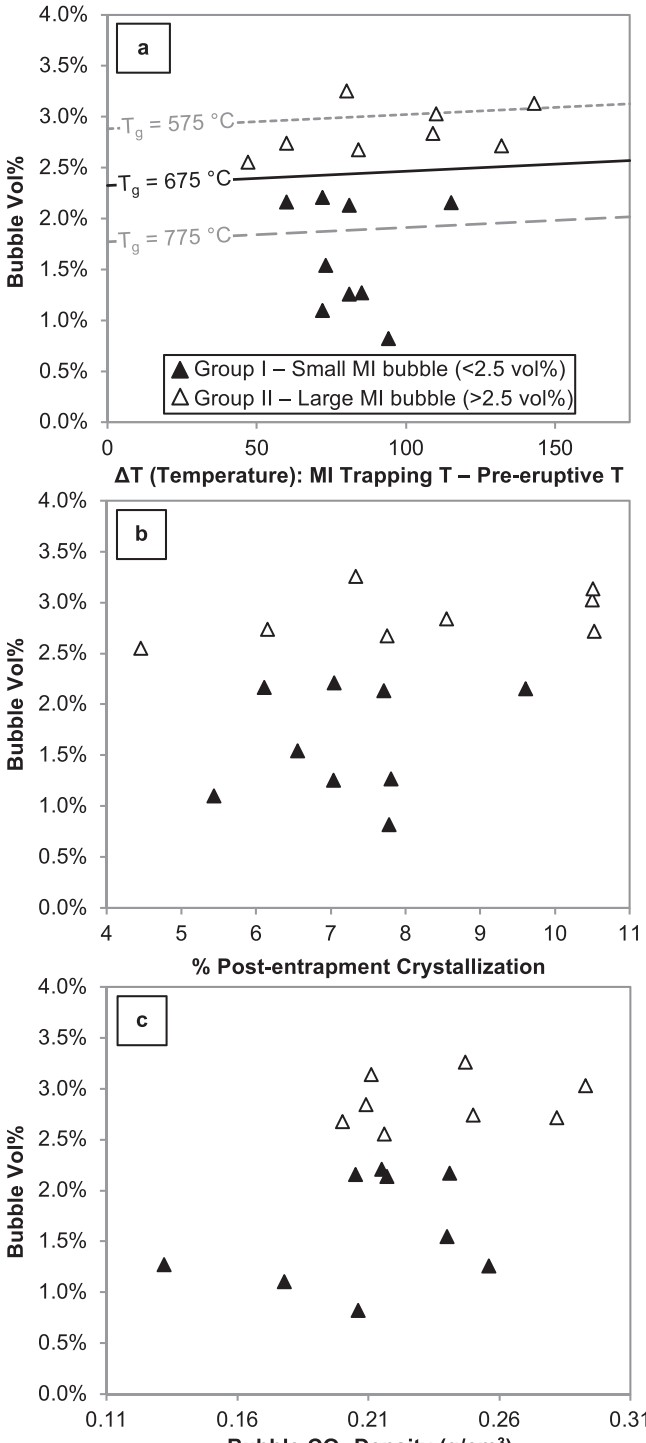

**Fig. 3 Bubble vol% vs. MI properties to illustrate variable formation mechanisms for the two groups.** Filled symbols are MIs with bubbles <2.5 vol% (Group I) and open symbols are MIs with bubbles >2.5 vol% (Group II). **a** Results from modeling post-entrapment bubble formation. The solid line indicates the maximum bubble size (vol%) attainable from post-entrapment crystallization and shrinkage assuming a glass transition temperature of 675 °C. **b** Percentage of post-entrapment crystallization experienced by each MI (see "Methods"). **c** Bubble $CO_2$ density ($g/cm^3$) as determined by Raman spectroscopy. Error in MI bubble vol% is 15% (i.e., 0.3 vol% error for a 2 vol% bubble); see "Methods" for details on error estimates.

is a function of the difference between the temperature of the magma when the MI is trapped and its temperature just prior to eruption ($\Delta T$). The additional early-stage bubble volume generated during post-entrapment crystallization is determined by the amount of crystallization that occurs, which also depends on $\Delta T$. Modeling of late-stage shrinkage requires an estimation of the glass transition temperature ($T_g$), which varies based on total $H_2O$ content and quench rate[29].

The lines in Fig. 3a show the results of modeling the size of MI bubbles generated by both stages of post-entrapment cooling for the Sunset Crater magma. The value $\Delta T$ on the x-axis represents cooling prior to eruption and accounts for early-stage differential shrinkage and post-entrapment crystallization. The value of $T_g$ represents the post-eruptive quench temperature, while the lines represent the final bubble vol% resulting from combined early- and late-stage bubble growth for different values of $\Delta T$ and $T_g$. The results are strongly dependent on the value of $T_g$, which has been investigated experimentally in basaltic melts at cooling rates between 5 and 20 °C/min[29]. However, for a highly explosive eruption such as at Sunset Crater, the cooling of small clasts, and especially the free crystals analyzed here, is likely to occur much faster than 20 °C/min. Based on $T_g$ data[29] and the shape of the relaxation curve of a silicate liquid[30], we estimate $T_g$ for these MIs to be 675 °C (Fig. 3a, solid line). In order to illustrate the sensitivity of these estimates to $T_g$, we also plot modeled bubble volumes for values of $T_g$ at +/−100 °C from our preferred value (Fig. 3a, dashed lines). Based on these calculations, Group II MIs are too large to have formed from post-entrapment cooling alone.

However, an alternate explanation for the different bubble trends in Group I and Group II MIs is that they experienced different cooling histories either pre- or syn-eruption. We reject this hypothesis on the basis of our interpretation of eruption dynamics and associated cooling during subsurface ascent and syn-eruptive quench. First, the data suggest that the pre-eruptive (subsurface) cooling was not significantly different between the two groups of MIs. The MI compositions (see Supplementary Material) indicate that they originated from a batch of homogeneous magma at depth that ascended rapidly without any pause at shallower depths prior to eruption. Additionally, there is no correlation between the amount of post-entrapment crystallization experienced by the MIs and the bubble size (Fig. 3b) nor is there any difference in the amount of post-entrapment crystallization between the two groups of MIs. Second, syn-eruptive quench would have been rapid in a sub-Plinian eruption (>20 km high plume). However, if the rates of syn-eruptive quench differed among crystals, the bubble $CO_2$ densities would also show differences, because during quench the bubble grows without $CO_2$ diffusion into the bubble. In other words, MIs experiencing slower syn-eruptive cooling should have larger bubbles (i.e., Group II MIs) with lower bubble $CO_2$ densities, but this is not recorded in the bubble density data (Fig. 3c).

One additional factor that can affect the size of MI bubbles is $H^+$ diffusion out of the MI during pre-eruptive (subsurface) cooling. This process results in a lower partial molar volume of the MI, which can lead to contraction of the MI and formation of a bubble[31]. This process cannot be solely responsible for the differences in bubble sizes between Group I and Group II MIs given the similar $H_2O$ contents of all MIs (Fig. 2a) and that samples with nearly identical $H_2O$ contents have different bubble sizes. Two samples show relatively low $H_2O$ contents that could indicate some minor $H^+$ diffusion, but all other Group II MIs share similar $H_2O$ contents to Group I MIs and should not have been affected by $H^+$ diffusion.

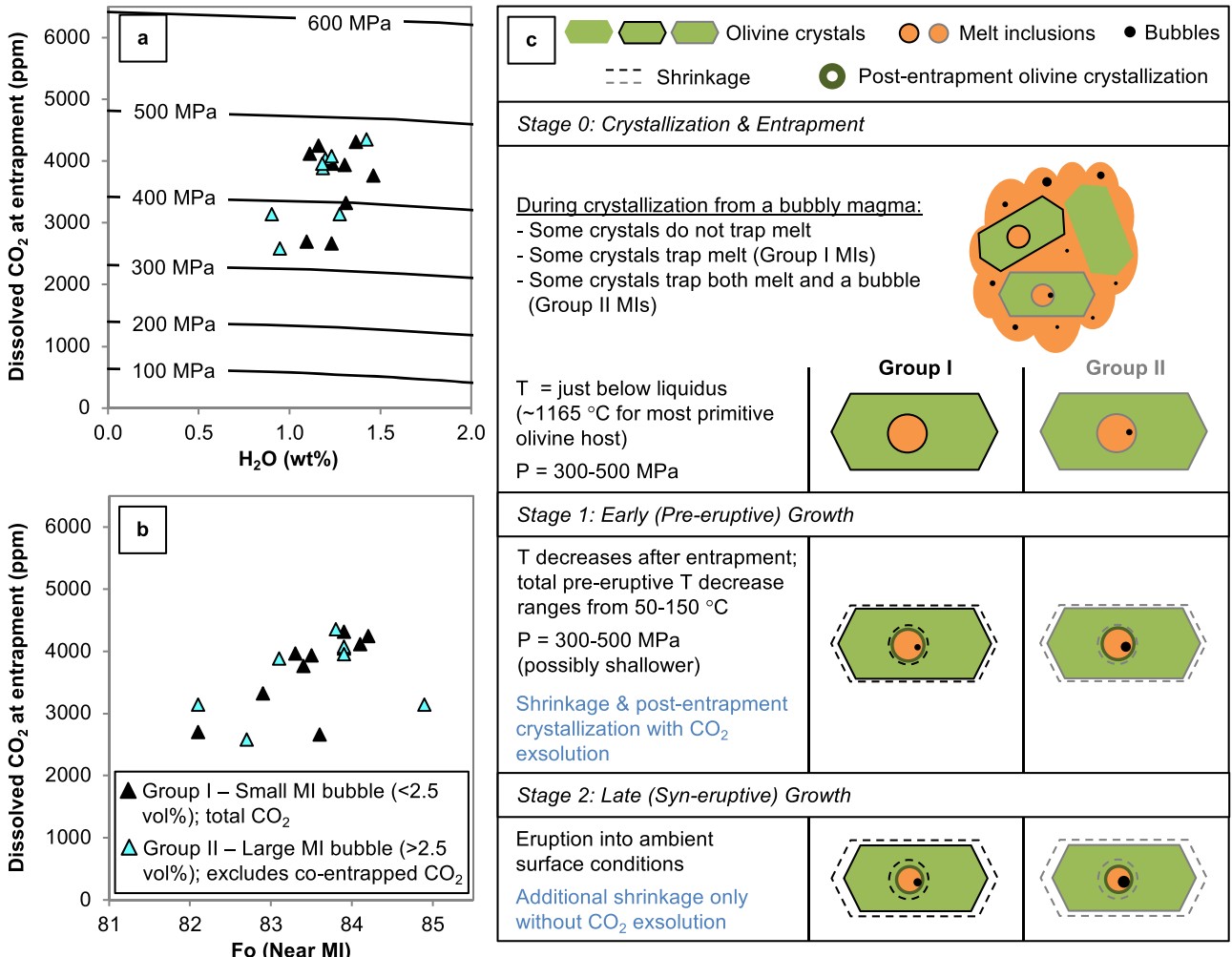

**Fig. 4 Conditions at MI entrapment and conceptual model of MI bubble growth. a, b** Dissolved $CO_2$ in the Sunset Crater magma at the time of MI entrapment. Group I samples (filled symbols) are plotted with their total $CO_2$ (glass + bubble). For Group II samples (cyan symbols), total $CO_2$ content is calculated assuming that 35% of the originally dissolved $CO_2$ content is sequestered into a bubble. **a** $H_2O–CO_2$ of MIs; curves are isobars from ref. [32]. **b** Dissolved $CO_2$ at entrapment vs. Fo content of the olivine host measured next to the MI. Error from FTIR analysis is 10% for $H_2O$ and 5% for $CO_2$, while the error in the total $CO_2$ content from bubble calculations is 5%. Error in MI bubble vol% is 15% (i.e., 0.3 vol% error for a 2 vol% bubble). See "Methods" for details on error estimates. **c** Interpretation of MI entrapment and bubble growth in MIs in the Sunset Crater magma. The modifications to the MIs shown are cumulative from the entrapment stage.

Given the lack of evidence for different cooling histories or significant $H^+$ diffusion, Group II MIs do have bubbles that are too large to have formed solely from post-entrapment cooling. Therefore, our favored explanation is that Group II MIs were trapped heterogeneously as two phases: liquid melt and exsolved $CO_2$. The Group II MIs have greater total $CO_2$ contents (Fig. 2a), as well as a greater percentage of the total MI $CO_2$ content in their bubbles (Fig. 2d), supporting the presence of an initial $CO_2$-rich exsolved phase.

We can estimate the original dissolved $CO_2$ content at entrapment for MIs with a co-entrapped exsolved phase by approximating the proportion of dissolved MI $CO_2$ that sequesters into Group II bubbles during post-entrapment processes. According to models[31], co-entrapped bubbles suppress exsolution of $CO_2$ from the melt because they counter some of the pressure loss due to shrinkage of the MI. The degree of suppression of $CO_2$ exsolution not only depends very strongly on the initial pressure but also on the initial bubble volume fraction and magma composition[31]. Using this model, we estimate that two MIs of alkali basalt composition at 450 MPa, one with no initial bubble and one with a 1 vol% bubble, should

lose nearly the same proportion of their original dissolved $CO_2$ to a bubble after 10% post-entrapment crystallization (~30 vs. 26%). Thus the proportion of originally dissolved $CO_2$ that exsolves into Group II bubbles should be similar to that of Group I bubbles (~35% on average) as the two groups experienced similar amounts of post-entrapment crystallization (Fig. 3b).

If we reconstruct the dissolved volatile contents at MI entrapment, assuming Group II MIs lost 35% of the originally dissolved $CO_2$ to a bubble, the two groups of MIs overlap in dissolved volatile content at entrapment (Fig. 4a). Note that the two Group II samples with the lowest $H_2O$ contents could be affected by $H^+$ loss via diffusion as discussed above. $H^+$ loss would facilitate additional $CO_2$ exsolution during pre-eruptive cooling, thus the $CO_2$ contents at entrapment shown in Fig. 4a may be slightly underestimated for these two samples. The reconstructed $CO_2$ at MI entrapment also clarifies the crystallization process, yielding one primary trend with Fo that suggests simple exsolution of $CO_2$ from the magma as crystallization proceeds (Fig. 4b).

Based on these estimates of the original dissolved volatile content, Group II MIs were trapped with 487 to 2226 ppm of exsolved $CO_2$

in a bubble. Assuming that these heterogeneously entrapped bubbles formed as a result of oversaturation in a magma at 450 MPa and 1200 °C where the density of $CO_2$ is ~0.74 g/cm³, the bubbles would be ~9–18 μm in diameter, corresponding to 0.18–0.80 vol% of their host MIs. Modeling indicates that these very small (<1 vol%) initial bubble sizes would not significantly suppress further $CO_2$ exsolution at these storage pressures[31], which validates our conclusion that both groups of MIs likely do lose a similar proportion of originally dissolved $CO_2$ to a bubble during early-stage (pre-eruptive) cooling.

**Implications for eruption style and scale.** The MI results show that the Sunset Crater magma was volatile saturated and included a $CO_2$-rich exsolved phase. Fluid-saturated isobars for Sunset Crater[32] indicate MI entrapment pressures between 300 and 500 MPa, corresponding to depths between ~12 and 17 km (Fig. 4a). The MIs with a co-entrapped exsolved phase (Group II; cyan symbols in Fig. 4a) show entrapment pressures consistent with this entire magma storage region, suggesting a bubbly magma deep in the system. Figure 4c presents the interpretation of MI entrapment and bubble growth in MIs in the Sunset Crater magma. Group I and II MIs are both trapped at storage depths between 300 and 500 MPa, but Group II MIs are trapped with a pre-existing bubble. The MIs next undergo early (pre-eruptive) bubble growth with $CO_2$ exsolution as a result of a slight decrease in temperature, causing both post-entrapment crystallization and shrinkage. Finally, the MIs are erupted and quenched, which yields late (syn-eruptive) bubble growth due to shrinkage only, and without further $CO_2$ exsolution. The modifications shown in Fig. 4c are cumulative from the entrapment stage. In summary, the MIs undergo the same two stages of bubble growth, but the bubbles in Group II samples remain proportionally larger due to their initial volumes.

Even if the Group II samples could be explained without heterogeneous entrapment, we would still expect $CO_2$ exsolution in this deep magma storage region on the basis of the total $CO_2$ results (Fig. 2a). As explained previously, the olivine data (Fig. 2b) show that the magma was volatile saturated throughout the entire depth range over which the MIs were trapped because samples follow a trend of decreasing $CO_2$ with decreasing Fo content (i.e., $CO_2$ exsolves as crystallization proceeds). If the total $CO_2$ in Group II MIs did represent the dissolved $CO_2$ at MI entrapment, the MIs would be trapped over a very wide pressure range from almost 600 to 300 MPa (i.e., Fig. 2a). A volatile-saturated magma of the Sunset Crater composition ascending from 600 to 300 MPa would have to exsolve nearly 3000 ppm of $CO_2$ into bubbles based on volatile solubility data[32].

Experimental work confirms that bubbles with similar dimensions to our estimates for the heterogeneously entrapped bubbles (i.e., 10s of micrometers in diameter) can be generated in alkali basalt magma at these pressures[33]. Bubbles with these properties (0.74 g/cm³ and ~20 μm diameter) would be coupled with the magma: bubble rise velocities would be on the order of $10^{-11}$ m/s based on calculations of the buoyancy force balanced against gravity and viscous forces, assuming Stokes drag and a range of reasonable magma viscosities, temperatures, and densities. This pre-eruptive system at Sunset Crater with a bubbly basaltic magma could be analogous to silicic magmas that produce highly explosive eruptions such as the Bishop Tuff rhyolite[34,35].

While Sunset Crater and silicic caldera-forming systems, exemplified by the middle-erupted Bishop Tuff[34,35], both may be characterized by eruption from bubbly magma storage zones, their storage depths and volatile compositions are notably different. Magma storage depths, calculated from MI saturation pressures, were 12–17 km at Sunset Crater but just 7 km for the Bishop Tuff magma[34,35]. The total dissolved $H_2O$ and $CO_2$

content of the Sunset Crater magma was 4.6 mol% at the time of MI entrapment, of which 89 mol% was $H_2O$ and 11 mol% was $CO_2$. In contrast, the Bishop Tuff magma contained significantly greater dissolved volatile content at MI entrapment (17 mol%), which was essentially 100 mol% $H_2O$[34,35]. The composition of the exsolved phase in equilibrium with each magma, as calculated from fluid isopleths (ref. [36] for rhyolite and ref. [32] for basalt), is also distinct. The exsolved phase in the Sunset Crater magma had only trace $H_2O$ (3 mol%) and was predominantly $CO_2$ (97 mol%). The Bishop Tuff, on the other hand, had an exsolved phase that was primarily $H_2O$ (92 mol% $H_2O$) with a small amount of $CO_2$ (8 mol%)[34,35].

Exsolved $H_2O$ is expected to play a large role in explosive eruptions, but an exsolved $CO_2$ phase at the greater depths of magma storage in basaltic eruptions may also produce the conditions necessary for large explosive eruptions. The significant overpressure from exsolved $CO_2$ can fracture wall rock[33,37], leading to rapid magma ascent and an explosive eruption. Given that storage zones of basaltic systems are expected to be deeper than those of silicic magmas based on neutral buoyancy considerations, and the greater solubility of $H_2O$ compared to $CO_2$, high $CO_2$ concentrations are required for an exsolved phase to exist at expected storage depths for basalt. This exsolved $CO_2$ phase may be necessary to initiate a pathway to the surface via overpressure and fracturing, and to drive rapid ascent from the deeper part of the system to the shallow region, where $H_2O$ exsolution is expected to take over the driving role. We thus propose that the exsolved $CO_2$ phase present at depth, as indicated by co-entrapped MI bubbles, was a necessary or critical condition that drove the sub-Plinian eruption of basalt at Sunset Crater, analogous to exsolution of $H_2O$-rich fluids driving caldera-forming silicic eruptions.

Rapid ascent due to an exsolved $CO_2$ phase at depth may be a common mechanism for mafic explosive eruptions. For example, Mt. Etna (Italy) has experienced some sub-Plinian and Plinian events and is similar in composition to Sunset Crater[38–40]. While some Etna MIs do contain significant dissolved $CO_2$[41], the $CO_2$ in Etna MI bubbles has not yet been quantified. Stromboli (Italy) is another mafic volcano that has produced explosive paroxysms, and exsolved $CO_2$ at depths up to 10 km beneath the crater has been proposed as the trigger for these events based on measured crater plume emissions[42]. Additionally, fluid inclusions in phenocrysts from Piton de la Fournaise (Réunion Island) suggest that $CO_2$ exsolution begins deep in the magma plumbing system of this volcano (500 MPa)[43]. Based on findings presented here, we would expect that the MI bubbles from the most explosive eruptions at these volcanoes might also contain significant $CO_2$ and evidence of a co-entrapped exsolved $CO_2$ phase, suggesting bubbly magma deep in the plumbing system. MI analysis following the procedures described here to assess MI bubble contents and formation would provide the data necessary to test this theory.

There are other processes, such as rapid shallow microlite crystallization and corresponding rheology changes, that have been proposed to explain explosive behavior in mafic magmas[4,5]. $CO_2$ exsolution was proposed as the trigger for the ~456 ka Pozzolane Rosse explosive mafic eruption at Colli Albani (Italy), but the $CO_2$ was not magmatic in origin, and the explosive nature was controlled by increased magma viscosity as a result of extensive decompression-induced leucite crystallization[44]. Extensive crystallization has also recently been proposed as the cause of the Masaya Triple Layer mafic Plinian eruption (2.1 ka; Nicaragua), which is relatively volatile poor[6]. While there is evidence in the Sunset Crater eruptive products for variable microlite crystallization in a portion of the groundmass, a significant fraction of the tephra in the sub-Plinian phases has a glassy and vesicular texture[1], suggesting that rapid microlite

crystallization cannot be the only important control. We further emphasize that rapid microlite crystallization is generally driven by exsolution of $H_2O$ from the magma in the shallow subsurface, and we therefore favor the idea that a deeper mechanism, such as exsolution of $CO_2$, is required to explain eruption initiation and rapid magma ascent in basaltic systems whose magmas ascend from greater depths to feed explosive eruptions.

**Atmospheric loading.** Sunset Crater erupted a significant volume of volatile-rich magma that reached the stratosphere during its most explosive phases. The eruption produced $0.52\,km^3$ dense-rock equivalent (DRE) ($2.8\,g/cm^3$ dense rock density) of volcanic material, of which $0.22\,km^3$ was erupted during the sub-Plinian phases[1]. As a result of the analysis presented here, along with these previously published volcanological characteristics, we estimate that the Sunset Crater eruption released ~0.6 Mt Cl, ~6 Mt $SO_2$, and ~11 Mt $CO_2$. $SO_2$ released during the sub-Plinian phases of the eruption (~2.45 Mt) could reach the stratosphere and generate $H_2SO_4$ aerosols. Assuming that 75% of the aerosol mass consists of $H_2SO_4$[45], the mass of stratospheric aerosols released was ~5 Mt. The remainder of the $SO_2$ was emitted during eruptive phases that reached the tropopause or upper tropo-sphere, at which levels it may also have induced atmospheric forcing[46]. These $SO_2$ values probably represent minima because they do not account for sulfur exsolved from un-erupted magma emitted during the eruption.

The atmospheric impact of explosive mafic eruptions, largely due to their high sulfur contents, may be comparable to their silicic counterparts. Previous interest in atmospheric effects of mafic eruptions has focused on large volume fissure eruptions, such as the Laki (Iceland) eruption of 1783–1784 (122 Mt $SO_2$)[11], which is a different eruptive mechanism entirely. Explosive silicic eruptions, although still much larger in terms of erupted volume, are better analogies to the dynamics of the Sunset Crater eruption. Two such historical eruptions, the 1991 eruption of dacite at Pinatubo (Philippines) and the 1815 eruption of trachyandesite at Tambora (Indonesia), resulted in profound atmospheric impacts. The Pinatubo eruption, which had significant impact on global climate for 3 years post-eruption, erupted 10 times the mass of magma ($5\,km^3$ DRE) as Sunset Crater ($0.5\,km^3$ DRE), but released just ~3 times the mass of $SO_2$ (17 Mt)[47]. The Tambora eruption was responsible for the "year without a summer," and while it erupted ~60 times the mass of magma ($30\,km^3$ DRE) as Sunset Crater, it released only ~9 times the mass of $SO_2$ (~55 Mt)[48]. While there is no evidence that Sunset Crater produced atmospheric impact similar to one of these large silicic explosive eruption located near the equator, its volatile output was significant.

Mafic scoria cones are the most common volcanic landform on Earth[49], but they are often overlooked because of their small stature. They are not well documented in the literature, partly because of poor preservation and burial by neighboring vents. But because of their high $CO_2$ and $SO_2$ contents, as well as their potential for sub-Plinian or larger eruptions, they can be important contributors of volcanic gases in the atmosphere. It is therefore possible that some unassigned events in the ice core record were derived from highly explosive mafic eruptions from scoria cone volcanoes.

## Methods

**Samples.** Free olivine crystals, generally coated in glass, were picked from a fraction of Sunset Crater tephra between 0.5 to 2 mm in diameter in size, mounted in epoxy, and examined for high-quality MIs for this study. The MIs were required to be glassy, ~>50 μm in diameter (often larger, depending on the placement of the bubble), away from any cracks or irregularities in the crystal, and tens of micrometers from the crystal rim. Approximately 20% of olivine crystals from Sunset Crater tephra contained viable MIs. MIs from Sunset Crater are commonly faceted,

ranging from ellipsoidal to negative crystal shapes. All MIs contain one bubble ranging in size from ~1 to 5 vol% in typical MIs. Rarely, multiple bubbles are present in a single MI, and in these cases, there is one primary bubble with the other bubble(s) being much smaller in size. We carefully selected primary MIs that did not exhibit evidence of extensive $H_2O$ loss or decrepitation. The only secondary modification physically apparent in the analyzed MIs was the bubble.

**$CO_2$ calibration.** To correct for instrument variability, any $CO_2$ densimeter[50] must be adjusted using $CO_2$ standards analyzed using a specific Raman instrument[51]. In this study, pure $CO_2$ gas was sealed in capillary tubes using a vacuum line to create a set of synthetic inclusions for use as standards[52]. These $CO_2$ gas standards have densities between 0.008 to $0.133\,g/cm^3$, which is consistent with the lower range of $CO_2$ densities measured in MI bubbles.

**Raman analysis.** MI bubbles were analyzed using Raman spectroscopic techniques. The Raman data were collected using a custom built Raman spectrometer in a 180° geometry at ASU in the LeRoy Eyring Center for Solid State Science (LE-CSSS). The sample was excited using a 150-mW Coherent Sapphire SF laser with a 532-nm laser wavelength. The laser power was controlled using a neutral density filter wheel and an initial laser power of 100 mW. The laser was focused onto the sample using a 50x super long working distance plan APO Mitutoyo objective with a numerical aperture of 0.42. The signal was discriminated from the laser excitation using a Kaiser laser band pass filter followed by an Ondax® SureBlock™ ultranarrow-band notch filter and a Semrock edge filter. The data were collected using a Shamrock 750 spectrometer from Andor® on an Andor iDUS back thinned Silicon CCD cooled to $-95\,°C$, and a $1200\,mm^{-1}$ grating was utilized to achieve optimal spectral resolution while preserving signal strength.

High-quality MIs were polished to <30 μm from the MI bubble and imaged with a petrographic microscope in preparation for Raman analysis. The MIs were specifically chosen to provide a representative sample of textural features observed in the Sunset Crater eruption (e.g., varying bubble volumes, MI sizes, MI shapes). In addition to these MIs and the four $CO_2$ standards, cyclohexane, naphthalene, and 1,4 bis (2-methylstyryl) benzene were also analyzed as Raman shift axis calibration standards during each Raman session. For the MI bubbles, the laser power at the sample was set to 6 mW (in isolated cases where the signal was low, the power was increased to 12 mW), and five 30-s scans were accumulated.

Raman spectra were first calibrated along the Raman shift axis using known values of the peaks of the three calibration standards (corresponding to 17 peaks in the measured range). Next, peak fitting was applied to the Fermi diad peaks for the $CO_2$ standards and MIs using a Gaussian–Lorentzian peak-fitting program, and preliminary $CO_2$ densities were calculated using the ref. [50] densimeter. A linear fit was obtained to adjust the Raman-calculated $CO_2$ densities of the capillary tube standards to their true densities, and all of the MI bubble densities were translated according to this fit.

The total contribution of $CO_2$ from the MI bubble was calculated from the Raman data using a mass balance approach[16,18,24]. In addition to the bubble $CO_2$ density, which is determined by the Raman analysis, this calculation requires density of the MI glass and the $CO_2$ concentration of the glass, as well as the volumes of the MI glass and bubble. The $CO_2$ contents of the MI glasses were determined by FTIR, which is described in the "Dissolved volatile analysis" section below. The densities of the MI glasses were determined from major element composition and $H_2O$ content, and this calculation is also described in the "Dissolved volatile analysis" section below. Volumes were determined from photomicrographs of the MIs, first using ImageJ to trace the area of the MI. This area was fit to an ellipse using the software, and volumes were calculated assuming the MI is an ellipsoid with a third axis intermediate between the long and short axes of the fit ellipse. The volumes of MI bubbles were calculated assuming spherical geometry. The uncertainty in the volume of the MI is primarily due to the estimate of the third axis. The average difference in length between the long and short MI axes is ~17 μm, and so we assume that the error in the length of the third axis is ±10 μm. This yields ~5% error in the total $CO_2$ content (i.e., Fig. 2a), and ~15% error in the bubble vol% (i.e., ±0.3 vol% for a 2 vol% bubble).

The mass balance calculations to calculate the total $CO_2$ abundance of the MI were completed following these steps:

1. Calculate the mass of $CO_2$ in the MI glass: mass of the MI glass (glass density × glass volume) × $CO_2$ concentration of the glass;
2. Calculate the mass of $CO_2$ in the MI bubble: bubble density × bubble volume;
3. Calculate the mass fraction of $CO_2$ in the bubble: divide the mass of $CO_2$ in the bubble (#2) by the total mass of $CO_2$ in the MI (glass + bubble; #1 + #2);
4. Calculate the reconstructed (glass + bubble) $CO_2$ concentration: divide the $CO_2$ concentration in the MI glass by the value of one minus the mass fraction of $CO_2$ in the bubble (#3).

Raman results are listed in Supplementary Table 1.

**Major element analysis.** After Raman analysis, MI glasses were analyzed for major elements using a Cameca SX100 Ultra electron microprobe at the University of Arizona. Each element was counted for 20 s (10 s for Na) using 15 keV accelerating

voltage, 20 nA beam current, and a 15-μm spot size. Olivine compositions were measured at the same conditions using a focused beam. Compositions of naturally quenched inclusions studied by Raman were corrected for post-entrapment crystallization and Fe loss using Petrolog3[53]. These corrections were calculated using an oxidation state equal to the NNO buffer[54] and the olivine-melt equilibria model of ref. [55], which yields a Fe–Mg partitioning coefficient of ~0.3 at 1200 °C. The FeO$^T$ value of 11 wt% was selected from bulk rock data[1]. Corrected MI and olivine compositions are listed in Supplementary Table 1.

A second set of MIs from the same tephra sample was analyzed for sulfur and chlorine on the same electron microprobe instrument. These elements were counted for 180 s each using 15 keV accelerating voltage, 20 nA beam current, and a 15-μm spot size. S and Cl results are listed in Supplementary Table 2.

**Dissolved volatile analysis**. $H_2O$ and $CO_2$ contents of the MI glasses were determined by FTIR. MIs were doubly intersected and polished in preparation for transmission FTIR analysis. FTIR analyses were performed using a Nicolet iN10 MX instrument at the United States Geological Survey in Menlo Park. Spectra were collected between 5500 and 1000 cm$^{-1}$ wavenumber for 45 s (128 scans) with high spectral resolution, and a background was collected before analyzing each sample. The aperture was carefully maximized for each inclusion according to the size of the doubly exposed spot on the inclusion to obtain an optimal spectrum.

$H_2O$ and $CO_2$ contents were calculated using the Beer–Lambert Law:

$$C = \frac{MW * A}{\rho * \varepsilon * d} \tag{1}$$

where $C$ is concentration in wt%, MW is the molecular weight of the absorbing species, $A$ is the peak height (absorbance) of interest, $d$ is sample thickness in cm, $\rho$ is density of the sample in g/L, and $\varepsilon$ is a molar absorption coefficient in L/mol-cm. Absorbances ($A$) were measured after subtraction of French-curve baselines drawn beneath each peak to reproduce the spectra of volatile-free samples. Thicknesses were determined using a Zygo ZeScope optical profilometer in the LE-CSSS at ASU, which allowed for precise thickness across the FTIR aperture to be determined. Density was calculated for each MI using the method detailed in ref. [56] wherein molecular partial molar volume contributions are totaled for a dry glass and density is adjusted iteratively based on water content. For total water absorption at ~3500 cm$^{-1}$, the absorption coefficient used was 63 L/mol-cm from ref. [57]. In rare cases where a near-IR peak for OH at ~4500 cm$^{-1}$ was visible, the coefficient of 0.67 L/mol-cm from ref. [58] was used. The near-IR peak for molecular water at ~5200 cm$^{-1}$ was not able to be resolved in any of these spectra. In these basaltic glasses, $CO_2$ is stored in the melt as $CO_3$, and the absorption coefficient was calculated for the MI composition according to ref. [59], with an average value of ~320 L/mol-cm. The ref. [59] relationship was specifically calibrated for alkali-rich mafic magmas like Sunset Crater, and so we estimate ~5% uncertainty for $CO_2$ content, while the uncertainty in $H_2O$ content is ~10%.

Volatile contents were also corrected for olivine growth at the rim of the MI (post-entrapment crystallization) using the results of the major element corrections. $H_2O$, $CO_2$, and $K_2O$ display incompatible behavior with olivine crystallization. After calculation of volatile contents from FTIR spectra using Eq. 1, the ratio of analyzed and corrected $K_2O$ contents was used to adjust the volatile components for post-entrapment modification. FTIR results are listed in Supplementary Table 1.

**Bubble growth model**. We model post-entrapment MI bubble formation and growth following the method of ref. [20], employing differences in density to determine maximum bubble volumes. The pre-eruptive bubble volume is controlled by the difference in the temperature of the melt at the time of MI entrapment compared to its temperature just before the onset of eruption. This temperature difference is assessed during calculations of post-entrapment crystallization using Petrolog3[53]. The final bubble volume after the crystal and MI are quenched during eruption is determined from the density of the crystal host and melt at the glass transition temperature, which was discussed above and estimated to be ~675 °C.

First, post-entrapment crystallization of olivine is calculated in 35 °C temperature steps from the liquidus temperature to the minimum pre-eruptive temperature using rhyolite-MELTS version 1.2.0[60]. The liquidus temperature was calculated to be 1166 °C using rhyolite-MELTS. For all calculations in rhyolite-MELTS, we use a primitive (high MgO, low SiO$_2$) MI composition (sample a-06) to represent the initial melt composition. Volatile content, pressure, and oxidation state are also required inputs in rhyolite-MELTS. To calculate pressure, we use the highest dissolved $CO_2$ concentration measured by FTIR (~3100 ppm), and for $H_2O$, we take the highest $H_2O/K_2O$ ratio of the MIs multiplied by the $K_2O$ content of the primitive composition, yielding 1.51 wt% $H_2O$. We calculate the pressure for this volatile content to be ~387.5 MPa using a solubility model for the Sunset Crater composition from ref. [32]. An oxidation state equivalent to the NNO buffer was used for these calculations. Rhyolite-MELTS outputs the new melt composition, masses of liquid (melt) and olivine, and the olivine density at each temperature step.

Next, we determine the pre-eruptive bubble size based on differences in melt density. We use the results from rhyolite-MELTS and partial molar volume coefficients from refs. [61,62] to calculate the size of the cavity that forms as a result of crystallization of a higher-density olivine phase. This cavity volume must then be adjusted for shrinkage of the olivine host. The change in volume of the olivine host

is calculated using the volume at ambient temperature of 43.95 cm³/mol from ref. [63] and adjusted for temperature by thermal expansion coefficients from ref. [64].

Finally, the maximum bubble size that can form during eruption (quench) is calculated. The bubble grows only from the shrinkage process during quench, and so the final melt volume is determined solely from the melt density at the glass transition temperature. The shrinkage of the olivine host is again accounted for at this stage as described above. The bubble sizes calculated for both the pre-eruption and quench stages are added together, and a linear fit is determined for the data to display the relationship between bubble size fraction and $\Delta T$ (Fig. 3a). An example calculation of the model is shown in Supplementary Table 3.

## Data availability
All data acquired for this study are included in Supplementary Tables.

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

## Acknowledgements

This work was supported by the National Science Foundation grants EAR-1322078 and EAR-1642569 (to A.B.C. and K.R.), as well as a Geological Society of America Graduate Student Research Grant (to C.M.A.). We would like to extend a very special thanks to Matt Steele-MacInnis (University of Alberta) for discussions on Raman calibration and modeling volatile exsolution during post-entrapment crystallization. We also thank Jake Lowenstern (United States Geological Survey), Ken Domanik (University of Arizona), and Emmanuel Soignard (Arizona State University) for assistance with analytical instruments. We gratefully acknowledge the use of facilities with the LeRoy Eyring Center for Solid State Science at ASU.

## Author contributions

All the authors conceptualized the project, interpreted the results, and wrote the paper. C.M.A. performed the analyses and bubble volume model calculations.

## Competing interests

The authors declare no competing interests.
