## [Peer Review File · Nature Communications]

Reviewers' comments:

Reviewer #1 (Remarks to the Author):

Dear Dr. Allison and co-authors,
please find below my review of your submitted manuscript for consideration to Nature Communications.

GENERAL COMMENTS

The Authors present a study on Raman and FTIR melt (and bubbles) inclusions from Sunset crater eruption occurred at 1085 AD. The obtained data and the combination of MIs and bubble volatile contents allowed the Authors to re-calculate the total volatile budget of the eruption, proposing a method for the extrapolation of total volatile content when MI and bubble data is known. Based on this approach, the Authors argue that CO₂ contents within the MIs bubbles were high enough to justify the presence of CO₂-rich exsolved phases prior (or during initial) magma ascent. MIs recent studies largely demonstrated that post-entrapment CO₂ loss to the formed bubble can be as large as 90% of the total MI content and thus represent a first-order issue when dealing with total volatile budgets. Most of this previous literature is however obtained on low CO₂ contents samples. In this study the Authors present a new example on higher CO₂ pairs of MIs-bubbles, modeling the size of the bubble to obtain values of initial trapped volatile content (both as exsolved and dissolved in the melt).

The model is more than convincing, particularly on the discussion regarding other possible explanations (different cooling histories, H⁺ diffusion etc.), data quality is very high, methods are clear and well detailed. The work has the promise to be an interesting addition to the studies of MIs data, opening new scenarios in the interpretation of volatile budgets obtained so far. In this view, the manuscript well fits the standard of a high-quality journal as NAT COMM.

In its present form, however, the manuscript suffers from not being as general as the topic could allow. While the presented problem is broad-spectrum and the proposed model is convincing, the organization of the manuscript is too linked to the Sunset crater example only. This is my only concern about the paper, and my suggestion is a partial reorganization of the text in order to enhance the scientific impact for the community of the work.

1) Organization of the manuscript

As said above, my main concern about the work is how the Sunset crater data is used to frame the problem. While title, abstract, and introduction are general enough, the discussion section is too focused on the obtained data, leaving the reader with the feeling that the big step is yet to come. The comparison with Bishop Tuff goes in that direction, but could be improved. How this model could affect other available datasets? As I said, this approach opens important scenarios (what about Vanuatu, Etna, Stromboli, for which MIs data is available? Any chance to make inferences?) and this should be demonstrated more clearly. The last part on atmospheric impact is, to me, disconnected from the real topic and only dilutes the final message. I would suggest to delete it and recover space to broaden the first part of the discussion.

I believe the authors should provide additional information by comparing with published data to support the novelty and innovation of the work. I find this manuscript compelling and acceptable in NAT COMM; however, I believe it remains that the authors need to stress how this manuscript surmounts to a significant advancement in the field, the novelty or at least the impact needs to be sold more strongly, whilst not over-reaching. I still believe that the topic would definitely appeal to a broad scientific readership, especially to those

interested in volatile budgets and magma dynamics.

2) Crystals and MIs framework

Although the description of the methods is clear in the proper section, some important (if not fundamental) aspects are not presented in the work. I had a look at the previous Alfano et al. (2018) and Allison et al. (2019) works, in which other information on the Sunset crater eruption is provided. The first work mostly deals with physical volcanology data, and only basic petrographic and geochemical (whole-rock) data is presented. The second work presents H₂O-CO₂ experiments on mafic samples (including those from Sunset crater) by means of FTIR analyses. My suggestion is thus to include in this presented paper the general framework of the analyzed samples as a new technique (Raman) is used here. Basic geochemistry, and particularly mineral chemistry should be clearly shown to the reader. Are those samples homogeneous? What about the olivine crystals? Are they zoned or un-zoned? There is a large variability on MI's bearing olivine crystals (euhedral, skeletal etc.) which reflect different processes of nucleation and growth and the Authors should present a more detailed description of the selected material (providing description and photos and not only tables). This is important to reinforce the model and to allow colleagues to compare with other available cases.

Reviewer #2 (Remarks to the Author):

This paper proposes that the VEI 4 eruption at Sunset Crater was driven by, at least partially, a deeply sourced exsolved vapor phase that was predominantly CO₂. They provide evidence of two populations of vapor bubbles, which they use to reconstruct the pre-eruptive volatile contents of the melt inclusions. A comparison with large silicic eruptions is made and the authors suggest that large explosive eruptions like those at Sunset Crater could also potentially cause atmospheric forcing.

This paper provides a novel trigger mechanism for the VEI 4 eruption at Sunset Crater. Exsolved vapor phases have been discussed in more silicic systems, but there has been limited discussion of them in the literature. Moreover, the authors do an exceptional job discussing the melt inclusion corrections. From a citation standpoint, I can see this paper becoming well-cited based on this and the novel scenario.

Given all this, I think the paper is suitable for publication in Nature Communications after some moderate revisions.

1. The authors assume 36% correction for the Group II melt inclusions, based on the correction in Group I melt inclusions. The work by Steele-MacInnis et al shows that melt inclusions that trap bubbles don't lose as much CO₂ to the bubble as those melt inclusions that grow bubbles after entrapment. I suggest the authors

model the correction value based on the model in Steele-MacInnis et al. At the very minimum, if the authors keep the 36% value, then they need to indicate that these are maximum corrections.

2. The atmospheric forcing discussion is a bit of a stretch. The comparison volcanoes, Pinatubo and Tambora, were the same VEI or larger. In addition to their size, it was their location on Earth that affected atmospheric processes on a global scale. Sunset Crater was certainly a large eruption, which likely had a regional impact. However, I think it is too far north to affect global climate. Language needs to be added that acknowledges that Sunset Crater did not likely affect global climate, should a similar sized eruption occur at an equatorial monogenic cone, it seems plausible that volcanic forcing could occur.

Reviewer #3 (Remarks to the Author):

This is an interesting article that attempts at quantifying the volatile budget of the sub-plinian phase of the Sunset Crater eruption in Arizona, using new analyses and modelling of volatile contents in olivine hosted MIs.

The main claim is that deep mafic magmas erupting in large scale (sub) plinian eruptions are saturated at depth with a CO₂-rich phase, and that this deep gas is the key to eruption dynamics and to the potentially large impact on climate and environment. This is an important conclusion. That deep mafic magma is oversaturated with CO₂-rich gas is not a new idea overall, but the new evidence brought here, and the level of detail of bubble modelling, are probably the first one for a large-scale (sub-plinian) eruption.

The manuscript is well written and the modelling of post-entrapment volatile loss to bubbles is especially well done and illustrated. Although I think the manuscript is in overall good shape and may make a good contribution to Nat Comm., I still have some important concerns (listed below) that I think should be fully addressed in a revision. The following further evidence would be required to strengthen and widen the conclusions of the manuscript:

1. I see some dualism between aim of the manuscript and its content. The intent of the authors is to convince the reader that basaltic eruptions can produce significant environmental/climatic impact. This is fine, but then the manuscript spends most of its space discussing H₂O and CO₂ content and behaviour (they have no impact), while leaving poor space to S and Cl, the gases that count from an environmental perspective. S and Cl are not even mentioned in the results (lines 78 – 86), they are not illustrated in figures, and their abundances seem to have been measured in "other MIs from Phase 3 of the Sunset Crater eruption" see supplement. S and Cl jump in the story only at page 10, which is not acceptable. I urge the authors to discuss more in depth S and Cl systematics in their samples, including use of some figures. I also wonder why S and Cl have not been measured in same inclusions, as it seems to me to understand from the supplement's caption?

2. On a related point, the authors calculate the composition of the exsolved phase in equilibrium with the magma (lines 190-194), but this is limited to CO₂ and H₂O. Again, what about S and Cl, it should be possible to quantify S content in the deep gas using saturation models (solex, decompress) or experimentally derived Dv/m, which are available for S (and for Cl as well), although perhaps not at 300-500 MPa. I am saying this because the title "Volcanic gases in highly explosive basaltic eruptions" requires S content (and Cl, if possible) is quantified in the gas phase as well. One very unconstrained aspect of global CO₂ flux inventories (see Werner et al., 2019 in Carbon in earth book) is the CO₂/S ratio (and CO₂ flux) of explosive eruption gases. The authors have here an exceptionally good opportunity to contribute to this poorly known aspect, especially for mafic eruptions. It would be nice to know what the deep (reservoir) volcanic gas C/S ratio was, and what was the ratio during the eruption (I am assuming S loss during magma ascent in the conduit, and syn-eruptive degassing, although again these critical aspects are not touched)

3. There is limited description on the way eruption volatile budget is calculated (lines 218-221). More details are needed: a) are you using MI mean/max concentrations for each gas? B) are you considering matrix glass content (for S and Cl) in the budget? What is the uncertainty in your estimates? No mention is made on them in the article. There is very limited detail on volcanological features of the eruption, and on the related numbers (volume, density, vesicularity, crystallinity) that are required to calculate the volatile output. The reader is referred to Alfano et al., while I think a section is needed in the supplement or methods that discusses briefly the eruption and its deposits, including where the samples were taken.

4. The strongest point of the manuscript is modelling of post-entrapment gas loss to bubbles during cooling (lines 100-168). The authors make the convincing point that the contrasting bubble size and CO₂ contents of their two bubble populations is due to entrapment of pre-existing (not shrinkage) bubbles. This is great. I wonder however if the fraction and volume of pre-existing bubbles trapped in the Mis is truly representative of the pre-eruptive bubble CO₂ content in the reservoir. It is well possible that bubbles were heterogeneously distributed in the reservoir, so that the fraction of bubbles (and bubble CO₂) trapped in the forming inclusions was not representative of the reservoirs as whole, especially if bubbles are crystals were segregated by density contrast. In other words, the authors do an excellent job in modelling post-entrapment modification, but could do more in correcting/estimating pre-entrapment CO₂ loss to bubbles by using CO₂/Ba and CO₂/Nb ratios and Nb-Ba contents in the Mis (e.g., Hartley et al., 2014). I hope these data are available.

5. The presence of a CO₂-rich gas phase deep in the reservoir could be tested analysing for the presence of fluid inclusions. I wonder if FIs have been seen in these samples, and if so if they have been analysed for density and composition. Deeply trapped fluid inclusions are recurrent features of mafic magmas, so the authors should not oversell their evidence that deep mafic magma is oversaturated with CO₂. This is well known. I concur, however, that the new evidence brought here are probably the first one for a large-scale (sub-plinian) eruption.

Palermo, October 10
Sandro Aiuppa

Author responses in blue text

Reviewer #1 (Remarks to the Author):

GENERAL COMMENTS

The Authors present a study on Raman and FTIR melt (and bubbles) inclusions from Sunset crater eruption occurred at 1085 AD. The obtained data and the combination of MIs and bubble volatile contents allowed the Authors to re-calculate the total volatile budget of the eruption, proposing a method for the extrapolation of total volatile content when MI and bubble data is known. Based on this approach, the Authors argue that CO₂ contents within the MIs bubbles were high enough to justify the presence of CO₂-rich exsolved phases prior (or during initial) magma ascent. MIs recent studies largely demonstrated that post-entrapment CO₂ loss to the formed bubble can be as large as 90% of the total MI content and thus represent a first-order issue when dealing with total volatile budgets. Most of this previous literature is however obtained on low CO₂ contents samples. In this study the Authors present a new example on higher CO₂ pairs of MIs-bubbles, modeling the size of the bubble to obtain values of initial trapped volatile content (both as exsolved and dissolved in the melt).

The model is more than convincing, particularly on the discussion regarding other possible explanations (different cooling histories, H⁺ diffusion etc.), data quality is very high, methods are clear and well detailed. The work has the promise to be an interesting addition to the studies of MIs data, opening new scenarios in the interpretation of volatile budgets obtained so far. In this view, the manuscript well fits the standard of a high-quality journal as NAT COMM.

In its present form, however, the manuscript suffers from not being as general as the topic could allow. While the presented problem is broad-spectrum and the proposed model is convincing, the organization of the manuscript is too linked to the Sunset crater example only. This is my only concern about the paper, and my suggestion is a partial reorganization of the text in order to enhance the scientific impact for the community of the work.

1) Organization of the manuscript

As said above, my main concern about the work is how the Sunset crater data is used to frame the problem. While title, abstract, and introduction are general enough, the discussion section is too focused on the obtained data, leaving the reader with the feeling that the big step is yet to come. The comparison with Bishop Tuff goes in that direction, but could be improved. How this model could affect other available datasets? As I said, this approach opens important scenarios (what about Vanuatu, Etna, Stromboli, for which MIs data is available? Any chance to make inferences?) and this should be demonstrated more clearly. The last part on atmospheric impact is, to me, disconnected from the real topic and only dilutes the final message. I would suggest to delete it and recover space to broaden the first part of the discussion.

I believe the authors should provide additional information by comparing with published data to support the novelty and innovation of the work. I find this manuscript compelling and acceptable in NAT COMM; however, I believe it remains that the authors need to stress how this manuscript surmounts to a significant advancement in the field, the novelty or at least the impact needs to be sold more strongly, whilst not over-reaching. I still believe that the topic would definitely appeal to a broad scientific readership, especially to those interested in volatile budgets and magma dynamics.

We have reorganized the manuscript so that it is clear that the main point is the eruptive mechanism of exsolved CO₂ generating overpressure, with the possible climactic impact more appropriately discussed as a minor secondary point.

Owing to the novelty of analysis of melt inclusion bubbles, there have not yet been studies published on other explosive mafic eruptions where the total CO₂ content has been quantified. We speculate in the revised version of the manuscript that other systems of similar composition (Stromboli, Etna) with sub-Plinian events may also show evidence for exsolved CO₂ when their melt inclusion bubbles are assessed. We also discuss another mechanism (rapid microlite crystallization) recently proposed for explosive basaltic eruptions, but dismiss it as the single controlling factor.

2) Crystals and MIs framework

Although the description of the methods is clear in the proper section, some important (if not fundamental) aspects are not presented in the work. I had a look at the previous Alfano et al. (2018) and Allison et al. (2019) works, in which other information on the Sunset crater eruption is provided. The first work mostly deals with physical volcanology data, and only basic petrographic and geochemical (whole-rock) data is presented. The second work presents H₂O-CO₂ experiments on mafic samples (including those from Sunset crater) by means of FTIR analyses. My suggestion is thus to include in this presented paper the general framework of the analyzed samples as a new technique (Raman) is used here. Basic geochemistry, and particularly mineral chemistry should be clearly shown to the reader. Are those samples homogeneous? What about the olivine crystals? Are they zoned or un-zoned? There is a large variability on MI's bearing olivine crystals (euhedral, skeletal etc.) which reflect different processes of nucleation and growth and the Authors should present a more detailed description of the selected material (providing description and photos and not only tables). This is important to reinforce the model and to allow colleagues to compare with other available cases.

Complete geochemical data were previously available in the supplementary material, but we have added additional discussion of this data in the revised manuscript, including olivine compositions and sample homogeneity. We also include representative images of each of the two groups of melt inclusions. We have added further details regarding our calculations of the total volatile budget of the eruption (first paragraph of the “Atmospheric Loading” subsection).

Reviewer #2 (Remarks to the Author):

Summary of Allison et al – Volcanic gases in highly explosive basaltic eruptions

This paper proposes that the VEI 4 eruption at Sunset Crater was driven by, at least partially, a deeply sourced exsolved vapor phase that was predominantly CO₂. They provide evidence of two populations of vapor bubbles, which they use to reconstruct the pre-eruptive volatile contents of the melt inclusions. A comparison with large silicic eruptions is made and the authors suggest that large explosive eruptions like those at Sunset Crater could also potentially cause atmospheric forcing.

This paper provides a novel trigger mechanism for the VEI 4 eruption at Sunset Crater. **Exsolved vapor phases have been discussed in more silicic systems, but there has been limited discussion of them in the literature. Moreover, the authors do an exceptional job discussing the melt inclusion corrections. From a citation standpoint, I can see this paper becoming well-cited based on this and the novel scenario.**

Given all this, I think the paper is suitable for publication in Nature Communications after some moderate revisions.

1. The authors assume 36% correction for the Group II melt inclusions, based on the correction in Group I melt inclusions. The work by Steele-MacInnis et al shows that melt inclusions that trap bubbles don't lose as much CO₂ to the bubble as those melt inclusions that grow bubbles after entrapment. I suggest the authors model the correction value based on the model in Steele-MacInnis et al. At the very minimum, if the authors keep the 36% value, then they need to indicate that these are maximum corrections.

Using the Steele-MacInnis model, we calculated CO₂-loss to a bubble at conditions consistent with the Sunset Crater magma in samples with no heterogeneously entrapped bubble and samples with a 1 vol% bubble. The model suggests that these two types of MIs would lose nearly the same proportion of their originally dissolved CO₂ to a bubble. Based on these results and the calculated "excess" CO₂ in Group II bubbles, the heterogeneously entrapped bubbles would in fact be <1 vol%. We have added these details to the revised manuscript.

2. The atmospheric forcing discussion is a bit of a stretch. The comparison volcanoes, Pinatubo and Tambora, were the same VEI or larger. In addition to their size, it was their location on Earth that affected atmospheric processes on a global scale. Sunset Crater was certainly a large eruption, which likely had a regional impact. However, I think it is too far north to affect global climate. Language needs to be added that acknowledges that Sunset Crater did not likely affect global climate, should a similar sized eruption occur at an equatorial monogenic cone, it seems plausible that volcanic forcing could occur.

We have added the suggested language to emphasize that we are not arguing that the Sunset Crater eruption definitively affected global climate, but that scoria cones in general have the potential to impact the atmosphere under the right conditions.

The following are annotated comments from reviewer #2. Line numbers refer to the original manuscript (OM) and revised manuscript (RM).

Line 17 OM; Line 19 RM: Yet documented sounds a little weird in this sentence.

We are not aware of another scoria cone eruption of greater magnitude, but we do not want to assume that this eruption is the largest from a scoria cone in all of geologic history. There may have been other highly explosive scoria cone eruptions without preserved deposits, for example.

Line 75 OM; Line 81 RM: A brief section discussing the details of the eruption for context

We include more previously published details about the eruption throughout the revised manuscript, for example at the beginning of the “Atmospheric Loading” subsection.

Line 81 OM; Line 90 RM: There is no figure 1b

This was a typo in the original manuscript, but Figure 1 in the revised manuscript includes 3 panels.

Line 86 OM; Line 90 RM: Can you provide images of the melt inclusions?

We include representative images of each of the two groups of melt inclusions in Figure 1 of the revised manuscript.

Line 87 OM; Line 111 RM: Move the description of grouping to before your volatile content results because you have them in figure 2.

We have significantly reorganized the figures and the first few paragraphs of the “Melt Inclusion and Bubble Compositions” subsection, and the two different symbols are explained in the text when Figure 2 is referenced (i.e., Line 94 RM).

Lines 93-94 OM; Lines 118-121 RM: not a comment, but the reviewer suggested reorganization of this sentence

This sentence now reads: The two groups are also distinguished by the percentage of the total MI CO₂ that is contained in the bubble (Fig. 2d); most Group I MIs contain <40% of their total CO₂ content in their bubbles, whereas all Group II MI bubbles contain >40% of the total MI CO₂.

Line 132 OM; Line 173 RM: Call it out in the supplementary data

We have added a reference to the supplementary data at this point in the text.

Line 149 OM; Line 189 RM: This is oddly placed since you do not mention other samples before. If you keep it, please refer the reader to the supplementary data

We no longer mention specific sample numbers in the main text, and instead refer to the specific samples being discussed here as the melt inclusions with the lowest water content.

Line 162 OM; ~Lines 204-207 RM: This doesn't make sense. Group I bubbles sequestered 36% of the CO₂ because they formed after melt inclusion formation. Group II bubbles should necessarily sequester less because the bubble was there in the first place. Which is what you say in the very next sentence based on Steele-MacInnis et al. I think you should model the actual amount loss using the model in Steele-MacInnis for the Group II melt inclusions.

Using the Steele-MacInnis model, we calculated CO₂-loss to a bubble at conditions consistent with the Sunset Crater magma in samples with no heterogeneously entrapped

bubble and samples with a 1 vol% bubble. The model suggests that these two types of MIs would lose nearly the same proportion of their originally dissolved CO₂ to a bubble. Based on these results and the calculated “excess” CO₂ in Group II bubbles, the heterogeneously entrapped bubbles would in fact be <1 vol%. We have added these details to the revised manuscript.

Line 177 OM; Line 246 RM: It would be really interesting to compare the abundance of Group I vs Group II melt inclusions.

Our dataset is not large enough to make a compelling argument for the abundance of each type of melt inclusion (i.e., Group I vs. heterogeneously entrapped Group II). However, the data suggest that Group II melt inclusions were trapped throughout the storage region, so we do not expect them to be significantly more rare than Group I inclusions. We do explain in the revised text that the Group II inclusions are present throughout the entire magma storage region (lines 232-233 RM).

Line 178 OM; Line 250 RM: Is there any way to model the vol% of bubbles in the magma at depth and compare with the abundance of the different melt inclusion groups

In the final paragraph of the “Bubble Growth Modeling” subsection (lines 217-225 RM), we model the initial vol% of the heterogeneously entrapped bubbles. On the basis of these bubble properties, we additionally model the ascent velocities of these bubbles (lines 246-250 RM) to speculate on their behavior in the pre-eruptive magma.

Line 191 OM; Line 264 RM: A figure comparing the melt volatile content vs the vapor phase volatile content would be a good addition to help clarify this section.

We ultimately did not add a new figure to supplement this paragraph, but we will keep this in mind as an option if the reviewers do not think this point is clear in the revised text.

Line 202 OM; Line 273 RM: I wouldn't use excess phase because it may make readers think of excess sulfur. Better to refer to it as the vapor phase

We have changed any references of “excess vapor” to “exsolved phase” in the revised manuscript.

Line 208 OM; Line 280 RM: Again getting a handle on the vapor phase abundance would make this idea more robust.

We have calculated the pre-eruptive bubble properties in two paragraphs of the revised manuscript: lines 217-225 RM to determine the original size of the heterogeneously entrapped bubbles, and lines 246-250 RM to speculate on the behavior of bubbles with these properties in the pre-eruptive magma.

Line 214 OM; Line 285 RM: not a comment, but the reviewer suggested changing “has” to “have” in this sentence

This sentence is grammatically correct with “has”; it states: “the CO₂ [...] has not yet been quantified.”

Line 220 OM; Line 319 RM: How did you calculate these values?

We have added further details regarding our calculations of the total volatile output of the eruption in the first paragraph of the “Atmospheric Loading” subsection.

Line 230 OM; Line 329 RM: Did anyone do work on the 1886 Tarawera fissure eruption? If so, that would also be a good comparison to include. Also El Chichon 1982, although it was slightly more evolved.

El Chichon 1982 is more evolved (i.e., 55-60 wt% SiO₂) than the basaltic systems we are referring to here. To our knowledge, there are no published volatile data from samples from the 1886 Tarawera eruption.

Line 239 OM; Line 340 RM: The atmospheric effect of these eruptions was not just because of their size, but because of the volcanoes' location on earth. Equatorial volcanoes are more likely to affect atmospheric processes worldwide than higher latitude volcanoes. That's why Laki only affected Europe and North America, but not regions further away.

We have added the suggested language to emphasize that we are not arguing that the Sunset Crater eruption definitively affected global climate, but that scoria cones in general have the potential to impact the atmosphere under the right conditions.

Line 364 OM; Line 667 RM: Fig 1b?

The reference in the text to Fig. 1b was a typo in the original manuscript, but Figure 1 in the revised manuscript includes 3 panels.

Figure 1: There is no figure 1b.

The reference in the text to Fig. 1b was a typo in the original manuscript, but Figure 1 in the revised manuscript includes 3 panels.

Figure 2: I know you list this in the captions, but titles on these figures makes them more easily understandable.

We have significantly modified the figures in the revised manuscript, so the helpful titles suggested for the original figure 2 do not apply in the revised manuscript.

Figure 3: Is there any way to combine these plots? You could use color and different shapes for the groups. Right now it is kind of blah.

We ultimately did not change the symbol coloring in the figures, but we will keep this in mind as an option if the reviewers think this would help to communicate the main points of the paper.

After figure 4: You need one or two more figures. I mentioned one earlier in the text. But you definitely need a cartoon/schematic conceptually showing what you are suggesting. That CO₂ exsolves early producing a sufficient vapor phase develops, this triggers fracturing and magma ascent leading to the eruption.

We have significantly modified the figures in the revised manuscript, and so although the total number remains the same, there are additional panels in most of the figures.

Annotations in supplementary tables:

Supplementary Tables: Generally speaking, these tables are more useful if provided in .csv or .xlsx format.

In the original manuscript, the .xlsx file was automatically converted to .pdf. We were able to change the manner in which the supplementary data was uploaded for the revised submission, so they should be visible to reviewers (and eventually readers) as .xlsx files.

Supplementary Table 1: Are the uncorrected data provided somewhere else? If not, please provide those as well.

We did not include the uncorrected data in these tables, but plan to do so when the data are uploaded to a repository upon publication.

Supplementary Table 1: Extend the MI sample labels to each page for easier readability.

The .pdf files that were visible to the reviewers in the first submission required that long tables extended across multiple pages. Now that the reviewers can see the supplementary data in .xlsx format, it is not necessary to add repeated columns with the sample names.

Supplementary Table 1: Can you sort these by grouping for easier comparison?

The supplementary data is now available in .xlsx format, so the data can easily be re-ordered in the spreadsheet by bubble vol% to compare the two groups.

Supplementary Table 2: Why didn't you collect S, Cl on the same melt inclusions as your H₂O, CO₂ data? I understand that these data are still useful. However, when discussing volatile budgets wouldn't it be better to compare contents from the same melt inclusions.

The compositions of the melt inclusions, including S and Cl contents, are remarkably similar for all samples we have studied from this eruption, which is why did not measure S and Cl in the same melt inclusions for which we analyzed bubbles by Raman spectroscopy. The decision was made due to time and funding constraints.

Supplementary Table 3: I appreciate the detail of these footnotes.

We hope the footnotes provide enough detail to replicate this modeling procedure for future studies.

Reviewer #3 (Remarks to the Author):

This is an interesting article that attempts at quantifying the volatile budget of the sub-plinian phase of the Sunset Crater eruption in Arizona, using new analyses and modelling of volatile contents in olivine hosted Mis.

The main claim is that deep mafic magmas erupting in large scale (sub) plinian eruptions are saturated at depth with a CO₂-rich phase, and that this deep gas is the key to eruption dynamics and to the potentially large impact on climate and environment. This is an important conclusion. That deep mafic magma is oversaturated with CO₂-rich gas is not a new idea overall, but the new evidence brought here, and the level of detail of bubble modelling, are probably the first one for a large-scale (sub-plinian) eruption.

The manuscripts is well written and the modelling of post-entrapment volatile loss to bubbles is especially well done and illustrated. Although I think the manuscript is in overall good shape and may make a good contribution to Nat Comm., I still have some important concerns (listed below) that I think should be fully addressed in a revision. The following further evidence would be required to strengthen and widen the conclusions of the manuscript:

1. I see some dualism between aim of the manuscript and its content. The intent of the authors is to convince the reader that basaltic eruptions can produce significant environmental/climatic impact. This is fine, but then the manuscript spends most of its space discussing H₂O and CO₂ content and behaviour (they have no impact), while leaving poor space to S and Cl, the gases that count from an environmental perspective. S and Cl are not even mentioned in the results (lines 78 – 86), they are not illustrated in figures, and their abundances seem to have been measured in “other MIs from Phase 3 of the Sunset Crater eruption” see supplement. S and Cl jump in the story only at page 10, which is not acceptable. I urge the authors to discuss more in depth S and Cl systematics in their samples, including use of some figures. I also wonder why S and Cl have not been measured in same inclusions, as it seems to me to understand from the supplement's caption?

We have reorganized the manuscript so that it is clear that the main point is the eruptive mechanism of exsolved CO₂ generating overpressure and the possible climactic impact is more appropriately discussed as a minor secondary point. We did not measure S and Cl in the same melt inclusions for which we analyzed bubbles by Raman spectroscopy, but we did measure these species in a different set of MIs from this same phase of the eruption. These data are included in the supplementary material and discussed in the revised manuscript. The compositions of the melt inclusions, including S and Cl contents, are

remarkably similar for all samples we have studied from this eruption. We have explained this homogeneity more thoroughly in the revised manuscript.

2. On a related point, the authors calculate the composition of the exsolved phase in equilibrium with the magma (lines 190-194), but this is limited to CO₂ and H₂O. Again, what about S and Cl, it should be possible to quantify S content in the deep gas using saturation models (solex, decompress) or experimentally derived Dv/m, which are available for S (and for Cl as well), although perhaps not at 300-500 MPa. I am saying this because the title “Volcanic gases in highly explosive basaltic eruptions” requires S content (and Cl, if possible) is quantified in the gas phase as well. One very unconstrained aspect of global CO₂ flux inventories (see Werner et al., 2019 in Carbon in earth book) is the CO₂/S ratio (and CO₂ flux) of explosive eruption gases. The authors have here an exceptionally good opportunity to contribute to this poorly known aspect, especially for mafic eruptions. It would be nice to know what the deep (reservoir) volcanic gas C/S ratio was, and what was the ratio during the eruption (I am assuming S loss during magma ascent in the conduit, and syn-eruptive degassing, although again these critical aspects are not touched)

First, we have edited the title of the manuscript to more accurately reflect our focus on CO₂ in this eruption instead of volcanic gases in general. We do have S and Cl measurements from another subset of melt inclusions from this phase of the eruptions, and we discuss those numbers in the revised manuscript. Experimental work (Lesne et al. 2011 J.Pet.) show that S and Cl don't degas from alkali-rich basalts until much shallower levels (i.e., 100 MPa or shallower), so we expect that the concentrations we have measured in the glass portion of these melt inclusions represents the total S and Cl in the magma.

3. There is limited description on the way eruption volatile budget is calculated (lines 218-221). More details are needed: a) are you using MI mean/max concentrations for each gas? B) are you considering matrix glass content (for S and Cl) in the budget? What is the uncertainty in your estimates? No mention is made on them in the article. There is very limited detail on volcanological features of the eruption, and on the related numbers (volume, density, vesicularity, crystallinity) that are required to calculate the volatile output. The reader is referred to Alfano et al., while I think a section is needed in the supplement or methods that discusses briefly the eruption and its deposits, including where the samples were taken.

We have added further details regarding our calculations of the total volatile output of the eruption in the first paragraph of the “Atmospheric Loading” subsection.

4. **The strongest point of the manuscript is modelling of post-entrapment gas loss to bubbles during cooling (lines 100-168). The authors make the convincing point that the contrasting bubble size and CO₂ contents of their two bubble populations is due to entrapment of pre-existing (not shrinkage) bubbles. This is great.** I wonder however if the fraction and volume of pre-existing bubbles trapped in the MI is truly representative of the pre-eruptive bubble CO₂ content in the reservoir. It is well possible that bubbles were heterogeneously distributed in the reservoir, so that the fraction of bubbles (and bubble CO₂) trapped in the forming inclusions was not representative of the reservoir as whole, especially if bubbles are crystals were segregated by density contrast. In other words, the authors do an

excellent job in modelling post-entrapment modification, but could do more in correcting/estimating pre-entrapment CO₂ loss to bubbles by using CO₂/Ba and CO₂/Nb ratios and Nb-Ba contents in the Mis (e.g., Hartley et al., 2014). I hope these data are available.

We do not have trace element compositions for these samples, but accurate determination of fluid-mobile trace elements would likely require rehomogenization of the melt inclusions to resorb the exsolved phase. We have attempted rehomogenization of these samples using multiple methods, but have not been able to resorb the bubble even in experiments at high pressure. The fact that the “excess” CO₂ shows a wide range in these samples certainly suggests that bubbles at depth are not homogeneously distributed, and we have edited the language in the revised manuscript so that this is not implied. Furthermore, Figure 4a represents the relative depths of the crystals in this study, and clearly shows that both Group I and Group II crystals came from the same range of depths and therefore must represent a heterogeneously bubbly magma body.

5. The presence of a CO₂-rich gas phase deep in the reservoir could be tested analysing for the presence of fluid inclusions. I wonder if FIs have been seen in these samples, and if so if they have been analysed for density and composition. Deeply trapped fluid inclusions are recurrent features of mafic magmas, so the authors should not oversell their evidence that deep mafic magma is oversaturated with CO₂. This is well known. I concur, however, that the new evidence brought here are probably the first one for a large-scale (sub-plinian) eruption.

We emphasize here that we find evidence for exsolved CO₂ in the pre-eruptive magma storage region. Much of the literature on fluid inclusions in basaltic systems is focused at deeper levels (i.e., fluid inclusions in mantle xenoliths). However, Bureau et al. 1998 (Chem. Geol.) find fluid inclusions in phenocrysts from Piton de la Fournaise, and from their analysis speculate that CO₂ degassing occurs at similar pressures to the Sunset Crater system (500 MPa). We mention this instance of deep CO₂ exsolution in the revised manuscript. We have not found fluid inclusions similar to those in the Bureau et al. 1998 study in the Sunset Crater phenocrysts.

REVIEWER COMMENTS

Reviewer #1 (Remarks to the Author):

Dear Dr. Allison and co-authors,
please find below my review of your resubmitted manuscript for consideration to Nature Communications.

GENERAL COMMENTS

The manuscript is a reviewed version of a presented study on Raman and FTIR melt (and bubbles) inclusions from Sunset crater eruption occurred at 1085 AD that I have already reviewed. The obtained data and the combination of MIs and bubble volatile contents allowed the Authors to re-calculate the total volatile budget of the eruption, proposing a method for the extrapolation of total volatile content when MI and bubble data is known. Based on this approach, the Authors argue that CO₂ contents within the MIs bubbles were high enough to justify the presence of CO₂-rich exsolved phases prior (or during initial) magma ascent.

The model is well presented and solid, particularly on the discussion regarding other possible explanations for the presence of bubbles of various size (different cooling histories, H₂ diffusion etc.), data quality is very high, methods are clear and well detailed. The study is particularly valuable from a methodological point (CO₂ amounts in MI bubbles may be higher than previously thought, and this is the first investigation for CO₂ high-concentration magmas), and secondly because it opens new interpretations on magma ascent triggering mechanisms for basaltic, CO₂-rich melts. I agree that several analogues (i.e. Etna or Stromboli volcanoes) must be investigated in the near future with a similar approach.

This new version is largely improved, and the Authors did a great job in reorganizing it. Now it is well-balanced, and the final section on atmospheric loading has the right space.

The work will be an interesting addition to the studies of MIs data, opening new scenarios in the interpretation of volatile budgets obtained so far. In this view, the manuscript well fits the standard of a high-quality journal as NAT COMM.

There are a couple of minor issues that may be addressed. First, although methods are very-well detailed, I was curious to know if the authors made some comparisons among Raman and FTIR data on the same MIs (maybe I didn't get this point). This will reinforce data comparison with a comment on that (how Raman and FTIR "talk" each other).

There is a small part (lines 111-117) in which the two Groups are presented. I suggest to move this part before, as Figure 2 is already commented in the lines before.

Finally, Figure 1 is well-designed. I was wondering if the Authors may want to include a similar sketch as a final cartoon in which they show how their model works (magma + exsolved CO₂ being trapped by olivines at a first stage, and Group I trapped at a later stage). A simple sketch would be an added, fast-reading way to deliver the message, but if there is space and appetite.

I would also like to apologize for my delay.

Reviewer #2 (Remarks to the Author):

This is the second time I have reviewed CO₂ exsolution driving highly explosive basaltic eruptions by Allison et al.

The authors have provided thorough and thoughtful responses to the reviewer comments. As such, the text is improved and the arguments are clear to the reader.

For the most part, the figures are easier to understand.

I would like to see modification to the symbols in Figure 4. Figure 4 shows the reconstructed CO₂ values for the melt inclusions, but uses the same symbols as those in Fig 2, 3 which show the original CO₂ contents. The authors should somehow distinguish these new CO₂ concentrations from the older CO₂ concentrations shown in Figs 2, 3. This could be completed with different symbols, variations in shading/color, etc.

Dawn C.S. Ruth
USGS

Reviewer #3 (Remarks to the Author):

I have read the revised manuscript and I am satisfied by the revisions made. The revised manuscript is now more focussed on the main findings (the role on CO₂ on eruptive processes), while less space/emphasis is given to the climatic implications. I am now happy to support publication on Nat Comm

Sandro Aiuppa

Author responses in blue text

Reviewer #1 (Remarks to the Author):

Dear Dr. Allison and co-authors,
please find below my review of your resubmitted manuscript for consideration to Nature Communications.

GENERAL COMMENTS

The manuscript is a reviewed version of a presented study on Raman and FTIR melt (and bubbles) inclusions from Sunset crater eruption occurred at 1085 AD that I have already reviewed. The obtained data and the combination of MIs and bubble volatile contents allowed the Authors to re-calculate the total volatile budget of the eruption, proposing a method for the extrapolation of total volatile content when MI and bubble data is known. Based on this approach, the Authors argue that CO₂ contents within the MIs bubbles were high enough to justify the presence of CO₂-rich exsolved phases prior (or during initial) magma ascent. The model is well presented and solid, particularly on the discussion regarding other possible explanations for the presence of bubbles of various size (different cooling histories, H⁺ diffusion etc.), data quality is very high, methods are clear and well detailed. The study is particularly valuable from a methodological point (CO₂ amounts in MI bubbles may be higher than previously thought, and this is the first investigation for CO₂ high-concentration magmas), and secondly because it opens new interpretations on magma ascent triggering mechanisms for basaltic, CO₂-rich melts. I agree that several analogues (i.e. Etna or Stromboli volcanoes) must be investigated in the near future with a similar approach.

This new version is largely improved, and the Authors did a great job in reorganizing it. Now it is well-balanced, and the final section on atmospheric loading has the right space.

The work will be an interesting addition to the studies of MIs data, opening new scenarios in the interpretation of volatile budgets obtained so far. In this view, the manuscript well fits the standard of a high-quality journal as NAT COMM.

There are a couple of minor issues that may be addressed. First, although methods are very-well detailed, I was curious to know if the authors made some comparisons among Raman and FTIR data on the same MIs (maybe I didn't get this point). This will reinforce data comparison with a comment on that (how Raman and FTIR “talk” each other).

Raman and FTIR analyses were both performed on each melt inclusion, but on different parts of the melt inclusions. The Raman analyses assessed CO₂ density within the bubble only, while the FTIR analyses measured CO₂ (and H₂O) concentrations solely in the glass. To further clarify this is the text, we added a line at the beginning of the “Raman Analysis” subsection of the Methods: “MI bubbles were analyzed using Raman spectroscopy techniques.” We convert density to mass proportion following the steps listed in the Methods, and sum the results from each of these analyses to obtain the total volatile content of each melt inclusion. To compare the amount of CO₂ obtained from each of these analyses, we show proportion of the total MI CO₂ that is contained in the bubble in Fig. 2d. The CO₂ contents of the glass and bubble are also listed separately in Supplementary Table 1.

There is a small part (lines 111-117) in which the two Groups are presented. I suggest to move this part before, as Figure 2 is already commented in the lines before.

The reasoning for why we classify the MIs into two groups is provided throughout this entire subsection (“Melt Inclusion and Bubble Compositions”), using Figure 2 to illustrate the differences between the two groups. However, we agree that it could be confusing to show the two groups in figures before explaining them in the text. The two groups of MIs are first mentioned in the figures in Figure 1b-c. These panels are first referenced in the latest revision of the manuscript in line 83. We therefore define the groups at this point in the text, stating: “Throughout this section we present data demonstrating that MIs should be classified into two groups based on bubble vol%. MIs with bubbles <2.5 vol% are hereafter referred to as “Group I” (black filled symbols in Figs. 2-4; example MI in Fig. 1b) and those with bubbles >2.5 vol% as “Group II” (open symbols in Figs. 2-3; cyan symbols in Fig. 4a-b; example MI in Fig. 1c).”

Finally, Figure 1 is well-designed. I was wondering if the Authors may want to include a similar sketch as a final cartoon in which they show how their model works (magma + exsolved CO₂ being trapped by olivines at a first stage, and Group I trapped at a later stage). A simple sketch would be an added, fast-reading way to deliver the message, but if there is space and appetite.

We appreciate the suggestion to draw from Figure 1 to generate a sketch of how Group I and II MIs evolve from entrapment to final quench. We have included this new cartoon in the latest revision as Figure 4c. We hope this provides further clarity to our interpretation of this system.

Reviewer #2 (Remarks to the Author):

Dawn C.S. Ruth
USGS

This is the second time I have reviewed CO₂ exsolution driving highly explosive basaltic eruptions by Allison et al.

The authors have provided thorough and thoughtful responses to the reviewer comments. As such, the text is improved and the arguments are clear to the reader.

For the most part, the figures are easier to understand.

I would like to see modification to the symbols in Figure 4. Figure 4 shows the reconstructed CO₂ values for the melt inclusions, but uses the same symbols as those in Fig 2, 3 which show the original CO₂ contents. The authors should somehow distinguish these new CO₂ concentrations from the older CO₂ concentrations shown in Figs 2, 3. This could be completed with different symbols, variations in shading/color, etc.

This is an excellent suggestion to clarify how the data in Figure 4 differs from the CO₂ contents plotted in Figure 2. We have changed the symbol fill for Group II MIs in Fig. 4a-b to cyan, and indicated in the legend that this data excludes co-entrapped CO₂. The calculation of entrapped CO₂ for these samples is explained in the caption (as well as in the main text).

Reviewer #3 (Remarks to the Author):

I have read the revised manuscript and I am satisfied by the revisions made. The revised manuscript is now more focussed on the main findings (the role on CO₂ on eruptive processes), while less space/emphasis is given to the climatic implications. I am now happy to support publication on Nat Comm
Sandro Aiuppa

Reviewers' Comments:

Reviewer #1:

None

Reviewer #2:

Remarks to the Author:

This is the third time I have reviewed CO₂ exsolution driving highly explosive basaltic eruptions by Allison et al. I am pleased to see the authors made the last minor modifications to the figures. I support this manuscript for publication. Well done.

Dawn C.S. Ruth

Author responses in blue

Reviewer #2 (Remarks to the Author):

This is the third time I have reviewed CO₂ exsolution driving highly explosive basaltic eruptions by Allison et al. I am pleased to see the authors made the last minor modifications to the figures. I support this manuscript for publication. Well done.

Dawn C.S. Ruth

We agree that the suggestions for modifications to the figures have clarified our interpretations of the data. We thank Dr. Ruth and the other reviewers for their detailed reviews of this manuscript.